# A Promising Biomarker and Therapeutic Target in Patients with Advanced PDAC: The Stromal Protein βig-h3

**DOI:** 10.3390/jpm12040623

**Published:** 2022-04-12

**Authors:** Christelle de la Fouchardière, Pia Gamradt, Sylvie Chabaud, Maxime Raddaz, Ellen Blanc, Olivier Msika, Isabelle Treilleux, Sophie Bachy, Anne Cattey-Javouhey, Pierre Guibert, Matthieu Sarabi, Pauline Rochefort, Pamela Funk-Debleds, Clélia Coutzac, Isabelle Ray-Coquard, Patrice Peyrat, Pierre Meeus, Michel Rivoire, Aurélien Dupré, Ana Hennino

**Affiliations:** 1Cancer Research Center of Lyon, UMR INSERM 1052, CNRS 5286, F-69373 Lyon, France; christelle.delafouchardiere@lyon.unicancer.fr (C.d.l.F.); pia.gamradt@inserm.fr (P.G.); isabelle.treilleux@lyon.unicancer.fr (I.T.); sophie.bachy@inserm.fr (S.B.); matthieu.sarabi@lyon.unicancer.fr (M.S.); clelia.coutzac@lyon.unicancer.fr (C.C.); michel.rivoire@lyon.unicancer.fr (M.R.); aurelien.dupre@lyon.unicancer.fr (A.D.); 2Université Lyon 1, F-69000 Lyon, France; 3Centre Léon Bérard, F-69008 Lyon, France; sylvie.chabaud@lyon.unicancer.fr (S.C.); max.raddaz@gmail.com (M.R.); ellen.blanc@lyon.unicancer.fr (E.B.); olivier.msika@avec.fr (O.M.); anne.cattey-javouhey@lyon.unicancer.fr (A.C.-J.); pierre.guibert@lyon.unicancer.fr (P.G.); pauline.rochefort@lyon.unicancer.fr (P.R.); pamela.funk-debleds@lyon.unicancer.fr (P.F.-D.); isabelle.ray-coquard@lyon.unicancer.fr (I.R.-C.); patrice.peyrat@lyon.unicancer.fr (P.P.); pierre.meeus@lyon.unicancer.fr (P.M.)

**Keywords:** pancreatic adenocarcinoma, pancreatic cancer, BIGHPANC, TGFbeta, immune checkpoint inhibitors, prognostic analysis

## Abstract

With an overall survival rate of 2–9% at 5 years, pancreatic ductal adenocarcinoma (PDAC) is currently the fourth leading cause of cancer-related deaths in the industrialized world and is predicted to become the second by 2030. Owing to often late diagnosis and rare actionable molecular alterations, PDAC has not yet benefited from the recent therapeutic advances that immune checkpoint inhibitors (ICI) have provided in other cancer types, except in specific subgroups of patients presenting with tumors with high mutational burden (TMB) or microsatellite instability (MSI). The tumor microenvironment (TME) plays a substantial role in therapeutic resistance by facilitating immune evasion. An extracellular stromal protein, βig-h3/TGFβi, is involved in the pathogenesis of PDAC by hampering T cell activation and promoting stiffness of the TME. The study BIGHPANC included 41 patients with metastatic PDAC, and analyzed βig-h3 levels in serum and tumor samples to assess the βig-h3 prognostic value. βig-h3 serum levels are significantly associated with overall survival (HR 2.05, 95%CI 1.07–3.93; *p* = 0.0301). Our results suggest that βig-h3 serum levels may be considered a prognostic biomarker in patients with metastatic PDAC.

## 1. Introduction

Pancreatic adenocarcinoma (PDAC) incidence in 2020 accounted for 495,773 cases and mortality for 466,003 cases according to GLOBOCAN. While significant advances have been made in the treatment of other solid cancers during the last 20 years, the survival of PDAC patients remains low, with a 5-year survival rate of 10%. Owing to the absence of reliable diagnostic biomarkers or clinical symptoms, about 85% of PDACs are diagnosed at a metastatic or locally advanced stage, partly explaining low survival rates [1,2]. PDAC treatment is based on surgery when feasible and adjuvant chemotherapy. In stage IV, the treatment mostly relies on two different chemotherapy regimens: FOLFIRINOX and gemcitabine plus nab-paclitaxel [3,4]. Recently, olaparib has been approved in patients with germline *BRCA* mutation, as maintenance therapy after disease control with a first-line platinum-based chemotherapy regimen [5]. Contrary to other cancer types, immune checkpoint inhibitors (ICI) did not show improved outcomes in PDAC, except when high tumor mutational burden (TMB) or microsatellite instability (MSI) are evidenced [6,7]. The nature of the tumor microenvironment (TME) may prevent efficient immune responses [8]. Our group previously demonstrated that PDAC stromal cells showed a high secretion of 68 kDa protein βig-h3, also known as a transforming growth factor β-induced gene product [TGFβi], in the extracellular matrix (ECM), which severely curtails T cell activation [9,10] and macrophage polarization [11,12]. Furthermore, the use of a neutralizing antibody against βig-h3 in a pre-clinical mouse model of PDAC reduced the tumor growth by enhancing cytotoxic CD8^+^ T cell responses [10]. Since βig-h3 is not only secreted in the pancreatic tumor but also released in the patient’s serum, we aimed to study βig-h3 tumor and serum expression levels in PDAC patients. The study BIGHPANC (NCT03472716) initiated in May 2018, enrolling patients with PDAC from stage I to IV, aimed to measure βig-h3 tumor and serum expression levels and to determine the most appropriate conditions for βig-h3 quantitative rates in liquid biopsies. We present results issued from the recently completed cohort of patients with stage IV PDAC.

## 2. Material and Methods

### 2.1. Patients and Data Collection

The non-interventional prospective study BIGHPANC consecutively enrolled adult (≥18 years) patients with PDAC, from stage I to IV according to the Tumour Node Metastasis International Union against Cancer (TNM/IUCC) classification, treated at the Centre Léon Bérard (Lyon, France). Main inclusion criteria were: (1) histologically/cytologically proven PDAC and (2) no systematic treatment initiated. All patients provided a written informed consent. Then, a 5 mL blood sample and a tumor sample (surgical piece or tumor biopsy at initial diagnosis) were collected. In case of relapse, new samples of blood and tumor may be required. This prospective cohort planned to recruit 45 stage IV patients. The study received local approval from the ethics committee (Lyon Sud-Est IV) and declared with ID-IRCB number 2017-A03590-53. The study complied with the law of January 1978 (78-17) related to computing, data, and freedom known as ‘computing and freedoms’, and with rules 2016/679 of the European Parliament of 25 May 2018 concerning the general data protection regulation (GDPR) MR-004.

The study was initiated on 22 May 2018, and PDAC cohorts (stage I-III) are still open to recruitment; the stage IV PDAC cohort completed inclusions on 31 May 2020.

### 2.2. Evaluation Criteria

The primary objective of the study was to assess the correlation between the βig-h3 protein expression level in a tumor microenvironment and PDAC at the TNM/UICC stage. Secondary objectives included assessment of the correlation between the βig-h3 protein serum level and PDAC at the TNM/UICC stage, CD8^+^ T cell immune infiltrate, and overall survival.

### 2.3. Statistical Methods

Descriptive statistics were used to compare patient and tumor baseline characteristics between groups. Fisher’s exact test was performed for categorical data and *t*-test or Wilcoxon rank sum test for continuous data. *p* values < 0.05 were considered statistically significant. Overall survival was calculated from the date of initial diagnosis to the date of death or censored at the date of the latest news. Progression-free survival (PFS) was calculated from the date of initial diagnosis to the date of the first event, defined as progression or death from any cause or censored at the date of the latest news. Survival curves with associated log-rank tests were generated using the Kaplan–Meier method. Median follow-up was calculated using reverse Kaplan–Meier estimation. Univariate and multivariate Cox proportional hazards models were used to identify potential prognostic factors. Only sufficiently informative variables (showing less than 10% of missing data) with *p* values < 0.20 in the univariate analysis were used for the multivariate model. A stepwise backward selection using *p* =0.05 threshold was performed to generate the final multivariate model. Hazard ratios (HRs) are presented with 95% confidence intervals (CI). All statistical analyses were performed using SAS version 9.4 (SAS Institute Inc., Cary, NC, USA).

### 2.4. Biological Analysis Methods

Serum βig-h3 levels were determined by enzyme-linked immunosorbent assay (ELISA) kits (DY2935, R&D System) and analyzed using GraphPad Prism version 8 (GraphPad Software, San Diego, CA, USA). Immunohistochemistry analysis and phenotyping for tumor-infiltrating leucocytes were performed on four-micron-thick sections of formalin-fixed, paraffin-embedded (FFPE) PDAC tissue samples with antibodies against CD8 (clone SP57, Roche Ventana) and CD163 (10D6, Novacastra Leica). βig-h3 staining was performed using anti-TGFBI Atlas Ab (HPA017019, Sigma). The specimens were analyzed by an expert pathologist. All IHC procedures were performed in a fully closed platform (BenchMark XT, Ventana/Roche). Negative controls were performed using healthy tissue. CD8^+^ T cells were numbered as the mean value of counts in each of three representative high-power microscopic fields, representing approximately 1 mm^2^. The area of CD163 staining was determined by Fiji software in pixel units. The results are expressed in the pixel area of the whole biopsy section. Statistical analysis was performed using Student *t*-test with GraphPad Prism software. * *p*  <  0.05 were considered statistically significant.

## 3. Results

A total of 45 patients with stage IV metastatic PDAC were included. Forty-one patients had suitable samples for βig-h3 serum analysis and the median βig-h3 level (3957.7 ng/mL; range 214–31,143) was used as a cut-off to differentiate low (n = 21) and high (n = 20) βig-h3 serum level subgroups. Seven patients had available tumor samples. Missing samples mainly resulted from administrative issues since diagnoses were either performed in other hospitals (transfer issues) or only used cytological samples. Patient characteristics are summarized in Table 1. The median age was 65 years (range 41–78) and most of them were diagnosed with synchronous metastases, involving the liver (73.2%), the lymph nodes (39.0%), and the peritoneum (24.4%), with a median number of two metastatic sites (range 1–4). Furthermore, 80.5% of the patients were ECOG-PS 0–1. Only one patient had a history of pancreatic surgery for his primary tumor. Forty patients were treated with chemotherapy, with a median number of two lines of treatment (range 1–4) and FOLFIRINOX (72.5%) being the most frequently used regimen in the first-line setting.

### 3.1. βig-h3 Serum Levels Significantly Correlate with Overall Survival in Stage IV PDAC

With a median follow-up of 32.2 months (95%CI 32.2–37.5), the median overall survival of the entire stage IV cohort was 12.8 months (95%CI 8.4–14.8). Patients with low βig-h3 serum levels showed a significantly longer overall survival compared with the high βig-h3 serum level group (median OS: 14.8 months, 95%CI 9.7–21.3 versus 10.2 months 95%CI 3.0–13.1) (HR 2.05 1.07–3.93; *p* = 0.0301) (Figure 1, Table 2). Furthermore, the βig-h3 level remained a significant predictor for OS in a multivariate Cox regression model after adjusting for ECOG-PS and NLR (HR = 2.33 (95%CI 1.17–4.63) *p*= 0.0156) (Table 3). No significant correlation (*p* = 0.2577) between βig-h3 serum levels and PFS was detected.

### 3.2. βig-h3, CD8, and CD163 Staining in Tumor Biopsies

Tumor sections were used for βig-h3, CD8, and CD163 staining. IHC revealed that βig-h3 expression was fibrillar as previously reported [3] (Figure 2). The infiltration of CD8^+^ T cells and CD163^+^ macrophages is shown in Figure 2. In the high βig-h3 serum level group, we observed a significant increase in infiltrating CD8^+^ T cells compared to the low βig-h3 serum level group (Figure 2C). However, CD8^+^ T cells were trapped in the stromal compartment and were not in direct or close contact with the tumor lesions. In addition, we also noticed a trend toward a stronger CD163 macrophage recruitment in this group (Figure 2B). It is noteworthy that no correlation between βig-h3 serum levels and βig-h3 expression in the tumors was observed, since βig-h3 was expressed in all seven available samples from four patients with low βig-h3 serum levels and three patients with high βig-h3 serum levels.

## 4. Discussion

This report is the first to study the prognostic role of βig-h3 serum levels in PDAC stage IV patients and these promising results encourage further evaluation of βig-h3 as a non-invasive prognostic biomarker in this disease. Few available prognostic factors for stage IV PDAC are mainly clinical; these are represented by poor Karnofsky performance-status scores, synchronous metastases, hepatic metastases, low baseline albumin levels, and ages above 65 years [3,4]. Research on new biomarkers is critical for clinicians in order to assist in accurate cancer diagnoses at early stages, to better determine prognosis, and/or to appropriately select treatment and such contributive tools that would benefit patients with PDAC. Until now, carbohydrate antigen 19-9 (CA 19-9) has been the only biomarker routinely used in the clinic, mainly for its prognostic role in resectable PDAC [13]. However, its prognostic value has not been established and it cannot be considered a reliable independent prognostic biomarker, both in asymptomatic patients or patients with stage IV PDAC. Furthermore, CA 19-9 levels may increase in nonmalignant diseases or in other cancers, and may be absent in 5–10% of the patients who do not display the sialyl Lewis epitope; thus, there is room for improvement, both in early diagnosis and prognostic stratification [14]. Our previous experiments in mice showed that βig-h3 expression is induced by pancreatic intraepithelial neoplasias (PanINs) and is maintained in cancers [10]. The present study shows that βig-h3 serum levels significantly and independently correlate with overall survival in patients with stage IV PDAC. Our present results need to be confirmed in larger cohort of patient with stage IV PDAC in which the correlation between the [beta]ig-h3 tumor staining and serum level is to be explored, because tumor samples failed to be available for such experiment in this cohort owing to transfer/administrative constraints. It has been reported that in gestastional diabetes there was an increase of tgbi mRNA in the plasma [15]. Therefore, further investigation in correlation with other related physipathological states/diseases should be pursued.. Lastly, results from the ongoing stages I-III BIGHPANC cohorts will be of crucial importance to validate the increase in βig-h3 serum levels from stage I to stage IV. Furthermore, βig-h3 level could be assessed at diagnosis in PDAC patients at high-risk (i.e., patients >50 years with new onset of diabetes, patients who smoke tobacco, patients with known BRCA germline mutation, etc.).

Besides its role as a prognostic factor, βig-h3 could also present a therapeutic interest. Indeed, our data suggest the existence of an immunosuppressive TME in patients with high βig-h3 levels. Based on previous results in mice showing that in vivo βig-h3 depletion reduced tumor growth, this protein should be explored as a promising target in patients with PDAC. Our current work consequently focused in generating a humanized anti-βig-h3 Ab for testing in a clinical setting, to confirm the promising immunological role of stromal extracellular matrix protein βig-h3 in pancreatic cancer [16,17].

## 5. Conclusions

We demonstrate here that high βig-h3 serum levels are associated with poor prognosis in stage IV PDAC. Further investigation should shed light on the use of this marker as a companion test in therapy.

## Figures and Tables

**Figure 1 jpm-12-00623-f001:**
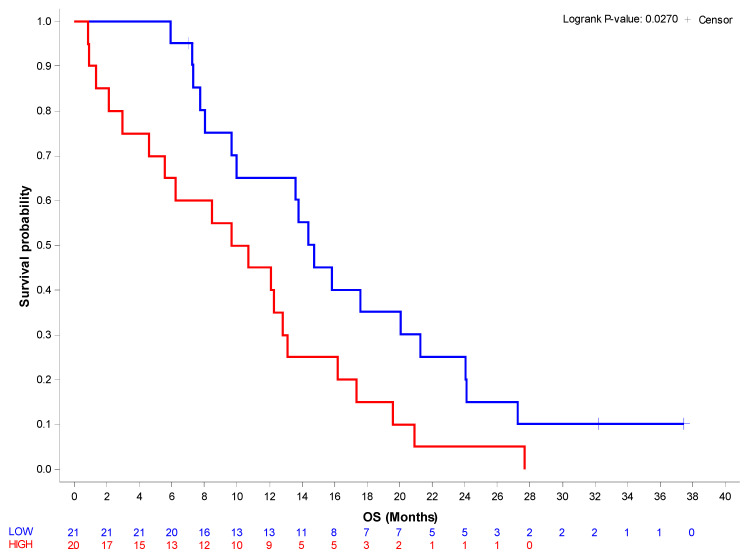
Overall survival according to βig-h3 serum levels. Overall survival was calculated from the date of initial diagnosis to the date of death or censored to the date of the latest news. Survival curves with associated log-rank tests were generated using the Kaplan–Meier method.

**Figure 2 jpm-12-00623-f002:**
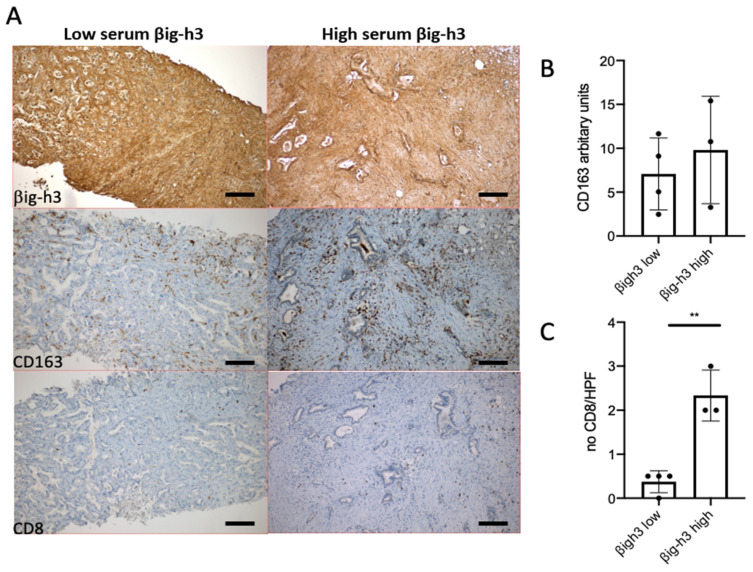
βig-h3, CD163, and CD8 staining in pancreas biopsies. (**A**) Immunohistochemistry staining of pancreatic biopsies for 2 patients with low and high βig-h3 serum levels stained for βig-h3, CD163, and CD8. Quantification of the CD163 staining area (**B**) of CD8^+^ T cells number (**C**) and in 7 patients. ** *p* < 0.01. HPF: high-power field. Scale bar 100 μm.

**Table 1 jpm-12-00623-t001:** Patient and tumor characteristics.

	All n = 41	βig-h3		Test
		Low (n = 21)	High (n = 20)	
Age				Wilcoxon
Median (min; max)	65.0 (41; 78)	61.0 (41; 76)	67.0 (45; 78)	*p* = 0.196
Gender				Fisher Exact *p* = 0.043
Female	13 (31.7%) 28 (68.3%)	10 (47.6%)	3 (15.0%)	
Male		11 (52.4%)	17 (85.0%)	
Performance Status (ECOG)				Fisher Exact *p* = 1.000
0	3 (7.3%)	1 (4.8%)	2 (10.0%)	
1	33 (80.5)	17 (81.0%)	16 (80.0%)	
2	5 (12.2%)	3 (14.3%)	2 (10.0%)	
Tobacco				Fisher Exact *p* = 0.758
Never	22 (53.7%) 19 (46.3%)	12 (57.1%)	10 (50.0%)	
Current/former		9 (42.9%)	10 (50.0%)	
Diabetes				Fisher Exact *p* = 0.697
No	34 (82.9%) 7 (17.1%)	18 (85.7%)	16 (80.0%)	
Yes		3 (14.3%)	4 (20.0%)	
Primary tumor localization				Fisher Exact *p* = 0.111
Head	14 (34.1%)	7 (33.3%)	7 (35.0%)	
Body	15 (36.6%)	5 (23.8%)	10 (50.0%)	
Tail	12 (29.3%)	9 (42.9%)	3 (15.0%)	
Metastatic site (at diagnosis)				Fisher Exact *p* = 0.484
Liver				
No	11 (26.8%)	7 (33.3%)	4 (20.0%)	
Yes	30 (73.2%)	14 (66.7%)	16 (80.0%)	
Lung				Fisher Exact *p* = 1.000
No	34 (82.9%) 7 (17.1%)	17 (81.0%)	17 (85.0%)	
Yes		4 (19.0%)	3 (15.0%)	
Peritoneum				Fisher Exact *p* = 0.277
No	31 (75.6%) 10 (24.4)	14 (66.7%)	17 (85.0%)	
Yes		7 (33.3%)	3 (15.0%)	
Lymph nodes				Fisher Exact *p* = 0.208
No	25 (61.0%) 16 (39.0%)	15 (71,4%)	10 (50%)	
Yes		6 (28.6%)	10 (50%)	
Bone				Fisher Exact *p* = 1.000
No	37 (90.2%) 4 (9.8%)	19 (90.5%)	18 (90.0%)	
Yes		2 (9.5%)	2 (10.0%)	
Number of metastatic sites				Wilcoxon *p* = 0.316
1	20 (48.8%)	13 (61.9%)	7 (35.0%)	
2	12 (29.3%)	3 (14.3%)	9 (45.0%)	
3	7 (17.1%)	3 (14.3%)	4 (20.0%)	
4	2 (4.9%)	2 (9.5%)	0 (0.0%)	
Median (min; max)	2.0 (1; 4)	1.0 (1; 4)	2.0 (1; 3)	
Differentiation grade				Fisher Exact *p* = 0.060
Unknown	2	0	2	
Low	8 (20.5%)	7 (33.3%)	1 (5.6%)	
Intermediate	24 (61.5%)	12 (57.1%)	12 (66.7%)	
High	7 (17.9%)	2 (9.5%)	5 (27.8%)	
Ca 19-9 at diagnosis (UI/l)				Wilcoxon *p* = 0.735
Median (min; max)	2497.0	2497.0	2027.0	
	(9; 128600)	(10; 128600)	(9; 84720)	
CEA at diagnosis (ng/mL)				Wilcoxon *p* = 0.666
Median (min; max)	8.0 (2; 810)	8.0 (2; 258)	8.5 (2; 810)	
Chemotherapy lines number				Fisher Exact *p* = 0.396
1	8 (20.0%)	2 (9.5%)	6 (31.6%)	
2	15 (37.5%)	8 (38.1%)	7 (36.8%)	
3	14 (35.0%)	9 (42.9%)	5 (26.3%)	
4	3 (7.5%)	2 (9.5%)	1 (5.3%)	
Chemotherapy regimen (L1)				Fisher Exact *p* = 1.000
FOLFIRINOX	29 (72.5%)	15 (71.4%)	14 (73.7%)	
Gemcitabine/nab-paclitaxel	1 (2.5%)	1 (4.8%)	0 (0.0%)	
Gemcitabine	3 (7.5%)	2 (9.5%)	1 (5.3%)	
Other	7 (17.5%)	3 (14.3%)	4 (21.1%)	

**Table 2 jpm-12-00623-t002:** Analysis for overall survival.

	Event/Total	Median (95% CI) ^KM^	Hazard Ratio (95% CI) ^Cox^	Survival Estimates (95% CI) ^KM^	*p*-Value
**βig-h3_cut-off**					0.0270 *
Low	18/21	14.8 (9.7–21.3)	Reference	6 months: 0.95 (0.71–0.99) 12 months: 0.65 (0.40–0.82) 24 months: 0.25 (0.09–0.45)	
High	20/20	10.2 (3.0–13.1)	2.05 (1.07–3.93)	6 months: 0.65 (0.40–0.82) 12 months: 0.45 (0.23–0.65) 24 months: 0.05 (0.00–0.21)	

^KM^ Kaplan-Meier method; ^Cox^ Cox model; * Logrank test.

**Table 3 jpm-12-00623-t003:** Uni- and multi-variate analysis for overall survival (Cox regression).

		Univariate Cox Model			Multivariate Cox Model		
		HR	CI95%	*p* Value	HR	CI95%	*p* Value
βig-h3 (cut-off median)	Low			0.0301			0.0156
	High	2.053	[1.07–3.93]		2.332	[1.174–4.633]	
ECOG-PS	0–1			0.1591			0.0413
	2,3,4	1.997	[0.76–5.23]		2.964	[1.044–8.418]	
Age	<65			0.6911			
	≥65	1.139	[0.60–2.16]				
Liver metastases	No			0.0561			ns
	Yes	2.093	[0.98–4.46]				
CA19-9 (cut-off median)	Low			0.3118			
	High	1.397	[0.73–2.67]				
Neutrophil-to-lymphocyte ratio	≤5			0.0016			0.0024
	>5	5.068	[1.85–13.9]		4.962	[1.764–13.964]	
Gender	F			0.0593			ns
	M	2.038	[0.97–4.27]				

## Data Availability

Data is contained within the article.

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
