# Peer review of "A Promising Biomarker and Therapeutic Target in Patients with Advanced PDAC: The Stromal Protein βig-h3"

_jpm, 2022, doi:10.3390/jpm12040623_

Round 1
Reviewer 1 Report
De La Fouchardière et al. report an interesting study on the stromal protein βig-h3 expression in metastatic pancreatic ductal adenocarcinoma.
The study is based on a homogeneous cohort of 45 patients and analyzes protein levels in relation to prognosis.
However some points should be better developed:
- The cut-off level of the biomarker was identified from the median value, would it not have been desirable to obtain it through roc curves?
- Are there other pathologies that could alter the value of this biomarker?
- How do the Authors think they can use this biomarker in a therapeutic way? Are there any studies or results about it?
- further stress the study limits .
Author Response
Reviewer 1
De La Fouchardière et al. report an interesting study on the stromal protein βig-h3 expression in metastatic pancreatic ductal adenocarcinoma.
The study is based on a homogeneous cohort of 45 patients and analyzes protein levels in relation to prognosis.
However some points should be better developed:
- The cut-off level of the biomarker was identified from the median value, would it not have been desirable to obtain it through roc curves?
In absence of assumptions concerning an hypothetic cut-off, median of the distribution has been used to split population into 2 subgroups (high/low levels) of equal size. This threshold is the one that offers the greatest statistical power. Using ROC curves to define a cut-off is questionable if the same sample is used for calibration and validation. Due to the small sample size (42 patients), we did not wish to share it in this way.
- Are there other pathologies that could alter the value of this biomarker?
We are not aware of other pathologies that can alter the value of the biomarker.
- How do the Authors think they can use this biomarker in a therapeutic way? Are there any studies or results about it?
We are currently investigating the correlation of the levels of big-h3 in patients before and after chemotherapy (stage IV) and also in stage I and II (before and after surgery).
- further stress the study limits .
We added in the discussion part that the study has limitations because of the small number of patients.
Reviewer 2 Report
The manuscript titled “A promising biomarker and therapeutic target in patient with advanced PDAC: the stromal protein βig-h3” submitted by Christelle de la Fouchardière et al., describes the study of prognostic biomarkers on 41 patients with metastatic PDAC following serum levels of stroma protein big-h3.
The topic is very interesting and the article is well written. I feel that the study is worthy of publication because it reports novel data and provides a comprehensive resource upon which future research can be based.
I have one observation that concerns the correlation between the protein big-h3 levels and related diseases. For example, is there a correlation between high protein levels and diabetes? I suggest the authors expand on this in the discussion section.
I also suggest including a paragraph of “Conclusion”.
Please review this sentence:
“Lastly, results from the ongoing stages I-III BIGHPANC cohorts will be of crucial im[1]portance to validate the results evidenced in mice. that the increase of big-h3 serum levels from stage I to stage IV could be assessed at diagnosis in PDAC patients at high-risk (>50 years with new onset of diabetes, tobacco smokers, known BRCA germline mutation etc).”

Author Response
The manuscript titled “A promising biomarker and therapeutic target in patient with advanced PDAC: the stromal protein βig-h3” submitted by Christelle de la Fouchardière et al., describes the study of prognostic biomarkers on 41 patients with metastatic PDAC following serum levels of stroma protein big-h3.
The topic is very interesting and the article is well written. I feel that the study is worthy of publication because it reports novel data and provides a comprehensive resource upon which future research can be based.
I have one observation that concerns the correlation between the protein big-h3 levels and related diseases. For example, is there a correlation between high protein levels and diabetes? I suggest the authors expand on this in the discussion section.
We are not aware of a study analysing the by ELISA the amount of big-h3 in the sera. Neverthess, it has been reported that in gestastional diabetes there was an increase of tgbi mRNA in the plasma (1). This information was added in the discussion.
I also suggest including a paragraph of “Conclusion”.
We added a paragraph of Conclusion.
Please review this sentence:
“Lastly, results from the ongoing stages I-III BIGHPANC cohorts will be of crucial im[1]portance to validate the results evidenced in mice. that the increase of big-h3 serum levels from stage I to stage IV could be assessed at diagnosis in PDAC patients at high-risk (>50 years with new onset of diabetes, tobacco smokers, known BRCA germline mutation etc).”
The sentence has been reviewed.
Lastly, results from the ongoing stages I-III BIGHPANC cohorts will be of crucial importance to validate the increase of big-h3 serum levels from stage I to stage IV. Furthermore, big-h3 level could be assessed at diagnosis in PDAC patients at high-risk (>50 years with new onset of diabetes, tobacco smokers, known BRCA germline mutation etc).
- Zhou H, Chen P, Dai F, & Wang J (2021) Up-regulation of TGFBI and TGFB2 in the plasma of gestational diabetes mellitus patients and its clinical significance. Ir J Med Sci.
